# Study on Oil Recovery Mechanism of Polymer-Surfactant Flooding Using X-ray Microtomography and Integral Geometry

**DOI:** 10.3390/molecules27238621

**Published:** 2022-12-06

**Authors:** Daigang Wang, Yang Song, Ping Wang, Guoyong Li, Wenjuan Niu, Yuzhe Shi, Liang Zhao

**Affiliations:** 1State Key Laboratory of Petroleum Resources and Prospecting, China University of Petroleum, Beijing 102249, China; 2China Petroleum Technology and Development Corporation, Beijing 100028, China; 3Jidong Oilfield of CNPC, Tangshan 063200, China; 4Research Institute of Petroleum Exploration and Development, PetroChina, Beijing 100083, China

**Keywords:** chemical flooding, pore-scale morphology, oil recovery mechanism, micro-CT, integral geometry

## Abstract

Understanding pore-scale morphology and distribution of remaining oil in pore space are of great importance to carry out in-depth tapping of oil potential. Taking two water-wet cores from a typical clastic reservoir in China as an example, X-ray CT imaging is conducted at different experimental stages of water flooding and polymer-surfactant (P-S) flooding by using a high-resolution X-ray microtomography. Based on X-ray micro-CT image processing, 3D visualization of rock microstructure and fluid distribution at the pore scale is achieved. The integral geometry newly developed is further introduced to characterize pore-scale morphology and distribution of remaining oil in pore space. The underlying mechanism of oil recovery by P-S flooding is further explored. The results show that the average diameter of oil droplets gradually decreases, and the topological connectivity becomes worse after water flooding and P-S flooding. Due to the synergistic effect of “1 + 1 > 2” between the strong sweep efficiency of surfactant and the enlarged swept volume of the polymer, oil droplets with a diameter larger than 124.58 μm can be gradually stripped out by the polymer-surfactant system, causing a more scattered distribution of oil droplets in pore spaces of the cores. The network-like oil clusters are still dominant when water flooding is continued to 98% of water cut, but the dominant pore-scale oil morphology has evolved from network-like to porous-type and isolated-type after P-S flooding, which can provide strong support for further oil recovery in the later stage of chemical flooding.

## 1. Introduction

Crude oil is one of the most important strategic materials in the world, which exerts a great impact on the national defense, economy, and energy security of a country. The contradiction between oil supply and domestic demand is becoming increasingly obvious, and the degree of dependence on the import of crude oil has exceeded 70% since 2020. Due to strong vertical and planar heterogeneity in chief oilfields such as Daqing and Shengli, high or extra-high water cut has been observed, showing a severely scattered distribution of remaining oil and the growing difficulty of stable oil production [1,2]. In order to achieve a higher oil recovery after water flooding, great efforts should be made to further improve microscopic oil displacement efficiency or enlarge swept oil volume in place. The subsurface crude oil can be effectively displaced by water-based injection by adding polymer, surfactant, and alkali in solution, as well as gas injection, including CO_2_, air, nitrogen, flue gas, etc. Several excellent technologies have been developed, such as water flooding, polymer flooding, polymer-surfactant flooding, and CO_2_ flooding. The Daqing oilfield in China has built the world’s largest production base of chemical flooding, which can further increase crude oil recovery of onshore clastic reservoirs by 10~20% after water flooding. Using alkali-surfactant-polymer (ASP) flooding, the final crude oil recovery can reach up to 60%. Chemical flooding has proven to be an efficient alternative to improve the unbalanced use of crude oil reserves and in-depth explore the potential of the remaining oil in the reservoir during the late stage of water flooding [3]. In order to develop novel technology for further enhanced oil recovery, it is significant to study the microscopic oil recovery mechanism of chemical flooding in strongly heterogeneous reservoirs.

The commonly used visual experimental devices mainly include microfluidic models [4,5,6,7], X-ray computerized tomography (CT) [8,9,10], focused ion beam-scanning electron microscopy (FIB-SEM) [11,12,13] and transmission electron microscope (TEM) [14,15]. The microfluidic models can visualize the fluid flow intuitively in pore space, which are widely utilized to reveal the percolation law of multiphase fluid. However, it is still restricted to 2-dimensional or 2.5-dimensional observation. In last decade, FIB-SEM and TEM imaging were rapidly developed and popularly used to quantitative characterization of rock microstructure at nano-micron scale in unconventional reservoirs, such as tight sandstone and shale. It suggests that both FIB-SEM and TEM are unable to accurately capture the dynamic evolution of fluid phase interface and multiphase fluid flow behaviors. The X-ray microtomography, for short, micro-CT, as an efficient nondestructive imaging tool, can obtain reliable gray images of rock samples with sub-micron resolution. The micro-CT has been widely involved in reservoir properties evaluation [16,17,18,19,20,21], multiphase fluid flow [22,23,24,25,26], enhanced oil recovery and CO_2_ geological storage [27,28,29,30,31,32].

Based on micro-focus CT imaging, Iglauer et al. [33] originally analyzed the difference in size distribution, morphology, and quantity of residual oil clusters in water-wet and oil-wet sandstones in the later stage of water flooding. The results show that when other conditions are similar, the residual oil in oil-wet sandstone is rough and flat, which is mainly attached to the pore surface or located in small pores, while the residual oil in water-wet sandstone exists in central positions of large pores. The size of residual oil in oil-wet sandstone is typically smaller than that in water-wet sandstone. Tanino and Blunt [34] conducted indoor experiments to investigate the underlying relationship between capillary trapping of residual oil and pore structure in sandstone and carbonate. It indicates that the residual oil saturation decreases with the increase in porosity and pore-throat ratio and increases as the coordination number of pore-throat reduces gradually. Lin et al. [35] and Bijeljic et al. [36] investigated the effect of micropore distribution of carbonate rocks on single-phase flow based on differential CT imaging and further estimated the connectivity between large pores. Gao et al. [37] (2019) subdivided the pores of water-wet carbonate rocks into macropores and micropores and analyzed the two-phase steady-state fluid flow in waterflooding by using differential CT imaging. The significant effect of movable oil in micropores on microscopic displacement efficiency and relative permeability under steady-state conditions is clearly clarified. Georgiadis et al. [38] studied the influence of injected pore volume of water on residual oil trapping in the process of displacement and imbibition. They concluded that regardless of imbibition and displacement, the increase in injection rate can improve the capillary number Ca, and the dissolution of rock microstructure at a high injection rate will enlarge porosity and permeability, thus leading to a reduction in residual oil saturation. Rücker et al. [39] and Pak et al. [40] investigated the two-phase fluid flow in sandstone and carbonate samples, respectively. When the capillary number Ca is higher than 10^−5^, the continuously distributed non-wetting oil clusters in pore space are fragmented into a series of isolated oil droplets. Reynolds et al. [41] explored the effect of low capillary number on the pore connectivity of Bentheimer sandstone based on micro-CT analysis. Zou et al. [42] further investigated the relationship between fluid relative permeability, spatial distribution, and pore connectivity in water-wet and mixed-wet sandstones. It can be inferred that, under mixed-wet conditions, a gradual increase in dynamic connectivity and pore-scale events of oil droplets can induce the non-equilibrium effect at the fluid interface, resulting in more energy dissipation during water flooding and lower effective permeability under the same water saturation. Spurin et al. [43] conducted micro-CT imaging to investigate the transition flow regime between micropores and macropores in carbonate rock samples. It concluded that increasing the displacement rate typically usually resulted in trade between two-phase fluids shifting toward small pores, and the transition flow regime can cause a snap-off of the displaced fluid, showing a significant influence on fluid relative permeability. 

In the last decade, a newly developed technique, integral geometry, has been introduced to characterize the pore-scale morphology and distribution of residual oil in the pore space of rock samples. Liu et al. [44] integrated the micro-focus CT imaging and the integral geometry to investigate the microscopic topology of two-phase fluids and its influence on relative permeability. The Minkowski function was originally used to characterize the microstructure and topological connectivity of the displaced fluid, revealing the remarkable effect of fluid microstructure and topological connectivity on relative permeability. Li et al. [45,46] carried out X-ray micro-CT imaging experiments in several water-wet sandstone samples with different scales of permeability and porosity. Based on digital image processing, shape factor, contact area, Euler coefficient, and other properties were proposed to characterize the pore-scale morphology and topological connectivity of residual oil in the pore space of rock samples. The water displacement process was divided into five stages: film flow, droplet flow, plug flow, porous flow, and cluster flow. It demonstrated that the cluster flow typically shows a higher relative permeability, while the relative permeability of the other flow patterns is relatively low. Khanamiri and Torsæter [47] conducted the imbibition experiments of injected water and surfactant for water-wet Berea sandstones. They applied synchronous CT imaging to capture the dynamic change of pore-scale fluid distribution and further investigated the influence of fluid topological connectivity on two-phase flow using integral geometry. McClure et al. [48], Bultreys et al. [49], and Miller et al. [50] proposed a geometric state function of capillary pressure based on quantitative analysis of fluid topology, which heavily depended on the curvature of pore structure, fluid saturation, contact area, and Euler coefficient. It is noted that numerous studies have focused on characterizing the pore-scale morphology and distribution of residual oil in rock samples during waterflooding and their influences on fluid flow, yet few comparative studies on pore-scale oil recovery mechanism of chemical flooding, especially the underlying difference of oil morphology and microstructure between waterflooding and chemical flooding, are performed.

To resolve these issues, two water-wet rock samples drilled from a typical clastic reservoir are selected as the research object. The X-ray micro-CT imaging is carried out to acquire high-resolution raw images of rock samples at different experimental stages of water flooding and P-S flooding. Using digital image processing, segmentation, and 3D visualization of rock microstructure and oil-water distribution at the pore scale is achieved. The dynamic change in pore-scale morphology and spatial distribution of residual oil under different displacement patterns are compared by using the integral geometry. The microscopic oil recovery mechanism of P-S flooding is ultimately revealed, which provides strong guidance for further tapping the oil potential in the later stage of chemical flooding.

## 2. Experiment 

### 2.1. Apparatus and Materials

The X-ray micro-CT imaging device used for chemical flooding experiments consists of four parts: a high-precision injection system, a flow simulation system, an X-ray micro-CT imaging system, and a separation and monitoring system of the produced fluid. The high-precision injection system is composed of a high-precision injection pump, a simulated oil container, and a simulated injected fluid container, which is adopted to real-time control the experiment conditions, including displacement rate, injected pore volume of displacing fluid, etc.). The flow simulation system denotes a rock sample that serves as a porous medium for oil-water two-phase flow. The X-ray computed tomography (CT) scanner manufactured by Bruker, Germany, is selected as the micro-CT imaging system, as shown in Figure 1. The high-energy micro-focus CT scanner is suitable for imaging high-density materials. It consists of four parts: an X-ray source, a flat-panel detector, a rotating carrier, and computer imaging equipment. It is utilized to acquire raw images of fluid distribution in pore space of rock samples during oil displacement experiments; the separation and monitoring system of produced fluid mainly includes a device for oil-water separation and measurement as well as an incubator. The produced fluid at the outlet of the rock sample can be accurately separated and monitored.

The experimental rock samples are two water-wet natural cores drilled from a typical clastic reservoir. The length and diameter of both samples are 5 cm and 5 mm, respectively. The smaller size of the natural cores, the higher-resolution raw images are obtained. Measured by traditional physical experiments, the porosities of the two water-wet cores were 0.371 and 0.247, respectively, and the gas permeabilities were 821.7% × 10^−3^ and 646.25 × 10^−3^ μm^2^, respectively.

The experimental oil is No. 26 industrial oil. At 25 °C, the density is 860 kg/m^3,^ and the viscosity is 54.3 mPa∙s. The simulated formation water is a 10% mass fraction of potassium iodide solution with a density of 1017.6 kg/m^3^ and a salinity of 14,250 mg/L. The polymer-surfactant binary oil-displacing system is compounded by AP-P5 polymer solution with a concentration of 1000 mg/L and the addition of CES-2 surfactant with a concentration of 0.5% in a ratio of 4:6. The viscosity of the polymer-surfactant solution is 15 mPa∙s.

During indoor experiments, the minimum displacement rate of a high-precision injection pump can reach 0.001 mL/min, and the monitoring accuracy of produced fluid is 0.01 mL, which fully meets the requirements of oil displacement experiments. The incubator can keep the produced fluid at a certain temperature to maintain the density and volume, thus ensuring measurement accuracy. The working voltage of the micro-focus CT scanner is 130 kV, the working current is 60 μA, and the exposure time is 1400 ms. True volumetric tomography can be achieved by rotating scanning with a rotation angle of 360° and a rotation step of 0.2°.

### 2.2. Experimental Procedure

In this study, a valve is controlled to realize the switch between the simulated oil and the displacing fluid. The rock samples are placed on the rotating carrier, and high-resolution micro-CT imaging of different cross-sections in rock samples can be achieved by 360° rotation of the carrier at an equal interval of 0.2°. The resolution of a single voxel was 13.84 μm. For the two rock samples, the experimental parameters of X-ray micro-focus CT imaging at different displacement stages were identical. The specific experimental procedures are as follows: 

(1) X-ray micro-CT scanning device is connected, installed, and tested. The nano-ray source is selected, the data acquisition software is unlocked, and the scanning parameters, including working voltage, current, and exposure time, are adjusted. Before indoor experiments, oil and salt washing are first conducted on the natural rock samples, and X-ray micro-CT imaging of the dry sample is then performed to investigate the rock microstructure; 

(2) The natural rock sample is vacuumed and saturated with water using a vacuum pump, and the saturation of simulated oil is achieved by switching the valve. When the core model reaches irreducible water saturation, X-ray micro-CT imaging is carried out, followed by real-time measurement of the produced fluid;

(3) Oil displacement experiments by injected water are performed at an injection rate of 0.01 mL/min, and X-ray micro-focus CT imaging of rock samples is accomplished when initial oil saturation and water flooding to 98% of water cut are achieved;

(4) The cleaned rock samples were repeatedly saturated with simulated formation water and oil. At the same injection rate, the P-S flooding experiments were carried out when the water cut reached up to 90%. X-ray micro-CT imaging of rock samples and measurement of produced fluid was completed at the time of initial oil saturation, water injection to 90% of water cut, and P-S flooding to 98% of water cut, respectively;

(5) Threshold segmentation and 3D reconstruction of fluid distribution in pore space of rock samples are recovered from the X-ray micro-CT raw images. Using integral geometry, pore-scale morphology and topological connectivity of residual oil before and after chemical flooding are systematically evaluated.

### 2.3. Core Flooding Results

After the X-ray micro-CT imaging-based experiments for water flooding and water injection followed by P-S flooding are accomplished, a total of 18,000 raw images of two water-wet rock samples can be obtained at 5 different experimental stages. The produced fluids are separated and measured, and the oil displacement efficiency at the end of water flooding and P-S flooding are calculated, as shown in Table 1.

As can be seen, compared with the result obtained when the water cut reaches 98% during water injection, a higher oil displacement efficiency can be observed when P-S flooding is continued to 98% of the water cut. The increased oil recovery is nearly 15%. It indicates that the polymer-surfactant binary displacing system can effectively mobilize pore-scale oil in the high water cut stage and significantly improve the development effect of the water-drive reservoir. When the same displacement pattern is utilized, the better the properties of the rock sample, the higher the oil displacement efficiency. Subsequently, the X-ray micro-CT raw images are processed to investigate the microstructure and topological connectivity of the displaced fluid using threshold segmentation, 3D reconstruction, and the newly developed integral geometry.

## 3. Image Processing and Data Analysis

### 3.1. X-ray Micro-CT Image Processing

In the study, AVIZO software is used to process the raw images of two water-wet rock samples obtained by X-ray micro-CT imaging at different experimental stages of chemical flooding, mainly including image pre-processing, threshold segmentation, 3D reconstruction, and visualization of pore-scale oil distribution. Various types of system noise can be discovered in the gray images of rock samples. Adjustments of image brightness and contrast, as well as sharpness, are firstly performed to improve the signal-to-noise ratio to make the raw images clearer, and reserve key features as much as possible. The threshold segmentation of CT raw images is crucial to characterize the pore-scale morphology and spatial distribution of the displaced fluid during chemical flooding. The commonly used segmentation methods include the Kriging thresholding algorithm, multi-threshold Otsu algorithm, and watershed algorithm [32,33]. The multi-threshold Otsu algorithm was utilized in this study. Since the density of the experimental oil (0.86 g/cm^3^) is close to that of deionized water (1.0 g/cm^3^), there typically exists some uncertainty when segmenting oil and water according to the difference of gray values between different phases. As mentioned before, we chose potassium iodide solution (1.0176 g/cm^3^) with a concentration of 10.0% instead of deionized water for X-ray micro-CT imaging. It can enlarge the gray difference between oil and water in raw images. In addition, the potassium iodide solution acts as a tracer, which can help to distinguish oil and water.

Figure 2 plots the segmented pore-scale oil distribution before and after water flooding and P-S flooding in the No. 2 rock sample. It shows that the content of microscopic oil in the pore space of water-wet cores gradually decreases as the displacement proceeds. When water flooding is continued to 98% of water cut, large quantities of scattered oil droplets are still unexploited. Compared with that of water flooding, the pore-scale oil droplets show a more scattered distribution when P-S flooding reaches 98% of the water cut. After the accurate threshold segmentation of raw images is completed, the rock skeleton of all scanned cross-sections and pore-scale fluid distribution at different experimental stages of water flooding and P-S flooding can be obtained. All these 2D segmented images are superimposed along the displacement direction using the marching cubes algorithm [22] to construct a 3D digital core model. The data elements with different colors indicate rock grain, oil, and water, respectively.

### 3.2. Geometric Analysis of Oil Droplets

In order to eliminate the effect of flow boundary effect as much as possible, it is necessary to crop a reasonable representative elementary volume (REV) from the established 3D digital core. Theoretically, the larger the size of the digital core, the more accurate the pore structure and physical properties of the rock samples, but a considerable demand for computer memory and computing power is also acquired. By considering the trade between computer memory and computing speed, the REV region used here is 256 × 256 × 256 voxels (3.543 mm × 3.543 mm × 3.543 mm). According to the 26-neighborhood principle, the data elements representing the non-wetting oil in the REV region are divided into disconnected oil droplets and numbered one by one. The geometric parameters for each oil droplet, including volume, surface area, equivalent diameter, perimeter, shape factor, and contact area with the grain surface at different displacement stages, are calculated.

The average volume of all disconnected oil droplets in REV of the water-wet rock samples at a certain displacement stage is described as:(1)$$\bar{V}=\frac{\sum_{i=1}^{N}V_{i}}{N}$$
where $\bar{V}$ is the average volume of a single oil droplet, μm^3^; N is the number of oil droplets; $V_{i}$ is the volume of a single oil droplet, μm^3^. The number of unexploited oil droplets implies the dispersion degree of oil at the pore scale. Affected by the synergistic effect of oil content and the number of residual oil droplets, the average volume of residual oil droplets is employed to reflect the microscopic oil-displacing effect of injected fluid.

Because of the rock microstructure and fluid injection history, the pore-scale distribution of residual oil droplets in the pore space of rock samples is very complicated. Four types of pore-scale oil morphology, as shown in Figure 3, are summarized as follows: ① Network-like. The residual oil droplets are distributed in several pore throats with a large volume and complicated microstructure; ② Porous-type. The residual oil droplets are located at relatively fewer pores and throats, and the shape is more complicated; ③ Isolated-type. The residual oil droplets are usually occupied in a single pore with a relatively regular shape; ④ Oil film-type. The residual oil droplets are mainly attached to the grain surface as oil film. Figure 3 displays the classification of pore-scale oil morphology in rock samples. The red color indicates the non-wetting oil phase, while the yellow color represents the pore space. For the quantitative criteria of pore-scale oil morphology, please refer to Wang et al. [18].

To further describe the dynamic change in residual oil microstructure during chemical flooding, the Euler coefficient [44] is introduced to evaluate the topological connectivity of non-wetting displaced fluid in pore space of the rock samples, which is defined as:
(2)$$\chi =\frac{M_{3}}{4\pi }=\beta_{0}-\beta_{1}+\beta_{2}$$
where χ is the Euler coefficient; $M_{3}$ is the integral Gaussian surface; $\beta_{0}$ is the number of isolated oil droplets in the REV region; $\beta_{1}$ is the number of pore throats associated with the network-like oil droplets; $\beta_{2}$ is the number of isolated water droplets surrounded by injected water, which can be neglected. A negative Euler coefficient indicates strong topological connectivity of oil microstructure, implying a network-like oil droplet; a positive Euler coefficient indicates poor topological connectivity, denoting an isolated oil droplet.

## 4. Oil Recovery Mechanism of Polymer-Surfactant Flooding

### 4.1. Mobilization of Oil Droplets by Injected Water

In this study, the No. 2 rock sample is used to demonstrate the mobilization mechanism of pore-scale oil in the process of water flooding. Figure 4 reflects the dynamic change in the oil content of different slices along the Z-upward direction. It can be seen that the residual oil droplets occupied at every slice have been effectively mobilized as water injection is carried out. However, the oil content within different slices of the water-wet rock sample differs greatly due to the microscopic heterogeneity of rock pore structure.

The dynamic changes in diameter, shape factor, and pore-scale morphology of residual oil droplets before and after water flooding are further compared. Figure 5 visualizes the divided results of residual oil droplets before and after water flooding in the No. 2 rock sample. Each colored block denotes a single oil droplet. Figure 6 depicts the diameter frequency distribution of residual oil droplets in the pore space of the No. 2 rock sample. It shows that the diameters of residual oil droplets before and after water flooding always obeys normal distribution. Due to continuous flushing of injected water, the average diameter of residual oil droplets shifts from 89.40 μm to 79.20 μm. The frequency distribution of residual oil droplets with diameters ranging between 69.21 μm and 124.58 μm gradually decreases. A sharp increase in the percentage of residual oil droplets with diameters ranging from 13.84 μm to 69.21 μm is observed, while the percentage of residual oil droplets larger than 124.58 μm almost remains unchanged. It implies that water injection mainly mobilizes the pore-scale oil droplets with diameters ranging from 69.21 μm to 124.58 μm, which are gradually stripped and broken into large quantities of smaller oil droplets as injected PV increases. The oil droplets with a diameter larger than 124.58 μm are difficult to be mobilized.

Figure 7 reflects the dynamic change in the pore-scale morphology of residual oil droplets within the REV region of the No. 2 rock sample before and after water flooding. It can be seen that as the injected pore volume of water increases, the pore-scale morphology of residual oil droplets in the pore space of the rock sample has significantly changed. The dominant pore-scale morphology of residual oil droplets before displacement is network-like, while the microscopic residual oil droplets after the displacement mainly show network-like and porous distribution. Table 2 summarizes the geometric properties of residual oil droplets in the pore space of the No. 2 rock sample. The average Euler coefficient of residual oil droplets shifts from −1974 to −535, indicating worse topological connectivity of residual oil droplets after waterflooding.

### 4.2. Pore-Scale Oil Recovery Mechanism by P-S Flooding

In order to investigate the pore-scale oil recovery mechanism by chemical flooding, X-ray micro-CT imaging of P-S flooding is performed on the two water-wet rock samples, which are recovered from different depths of a typical clastic reservoir. Thereafter, the digital image processing techniques and integral geometry are integrated to compare the pore-scale morphology and spatial distribution of residual oil droplets occupied in the pore space of rock samples under different experimental stages. The underlying pore-scale oil recovery mechanism by P-S flooding is finally explained.

Figure 8 and Figure 9 display the dynamic changes in the shape factor of residual oil droplets for the No. 1 and No. 2 rock samples after water flooding and P-S flooding. It indicates that only a small amount of complicated network-like oil droplets has been effectively mobilized after long-time flushing of injected water. There exists a large amount of oil droplets unexploited in pore space when water flooding reaches 98% of water cut. When P-S flooding is ended, most network-like oil droplets with diameters greater than 200 μm are successfully recovered. The result suggests that the polymer macromolecules during P-S flooding obviously improve the water-oil mobility ratio and reduce the water relative permeability, which can effectively enlarge the pore-scale swept volume of injected fluid. The addition of surfactant further reduces the oil-water interfacial tension, which greatly improves the flow capacity and displacement efficiency of the non-wetting oil phase. The synergistic effect of “1 + 1 > 2” between the strong oil-washing ability of surfactant and the enlarged swept volume of polymer makes more microscopic oil droplets recover.

Figure 10 and Figure 11 present the 3D visualized distribution of pore-scale residual oil droplets in both water-wet rock samples when P-S flooding reaches 98% of the water cut. Figure 12 displays the distribution histogram of pore-scale oil morphology in the two water-wet rock samples. The analysis shows that compared with the dominant network-like oil droplets after waterflooding, as shown in Figure 7, regardless of No. 1 and No. 2 rock samples, a large amount of network-like oil droplets can be effectively mobilized and recovered after P-S flooding. The major pore-scale morphologies of residual oil droplets have shifted from network-like to porous-type and isolated-type, which will be regarded as important targets for further oil potential tapping in the later stage of chemical flooding.

Figure 13 displays the frequency distribution of residual oil droplets’ diameters before and after P-S flooding in the two water-wet rock samples. The average geometric properties of residual oil droplets before and after P-S flooding are summarized in Table 3. It demonstrates that the average diameters of residual oil droplets decreased from 94.4 μm to 69.3 μm in the No. 1 rock sample and from 95.8 μm to 76.1 μm in the No. 2 rock sample. The proportion of small oil droplets with diameters smaller than 69.21 μm gradually rises, while the proportion of oil droplets with diameters greater than 124.58 μm decreases, indicating that the P-S displacement system mainly mobilizes the residual oil droplets with relatively large volumes. During P-S flooding, large network-like and porous-type oil droplets are continually broken into isolated oil droplets, resulting in a dramatic decrease in topological connectivity.

## 5. Conclusions

(1) Water flooding and P-S flooding experiments in two water-wet rock samples of a typical clastic reservoir were carried out, followed by X-ray micro-CT imaging at every experimental stage. The pore structure and oil-water distribution were segmented, and 3D reconstructed in 3D using digital image processing techniques. The pore-scale morphology and spatial distribution of residual oil droplets were further investigated using integral geometry to explore the oil recovery mechanism by P-S flooding.

(2) The core flooding experiments show that injected water mainly mobilizes oil droplets with diameters ranging from 69.21 μm to 124.58 μm, and oil droplets with diameters larger than 124.58 μm were difficult to be recovered. There still exists a large amount of residual oil droplets in the pore space of rock samples, mainly showing a network-like distribution.

(3) Compared with the water flooding, both the average diameters and topological connectivity of residual oil droplets in pore space obtained after P-S flooding will decrease dramatically, indicating a better oil displacement efficiency. Due to the synergistic effect of “1 + 1 > 2” between the strong displacement efficiency of surfactant and the expanding swept volume of polymer molecules, the oil droplets larger than 124.58 μm in diameter are gradually stripped. A more scattered distribution of pore-scale oil droplets has been observed when P-S flooding is continued to 98% of water cut. The dominant pore-scale morphologies of residual oil droplets shift from network-like to porous-type and isolated-type.

## Figures and Tables

**Figure 1 molecules-27-08621-f001:**
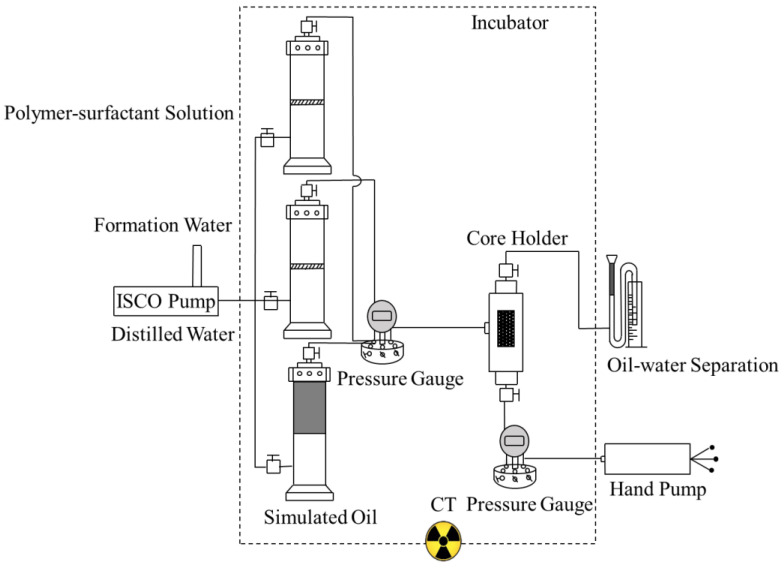
X-ray micro-focus CT imaging experimental device.

**Figure 2 molecules-27-08621-f002:**
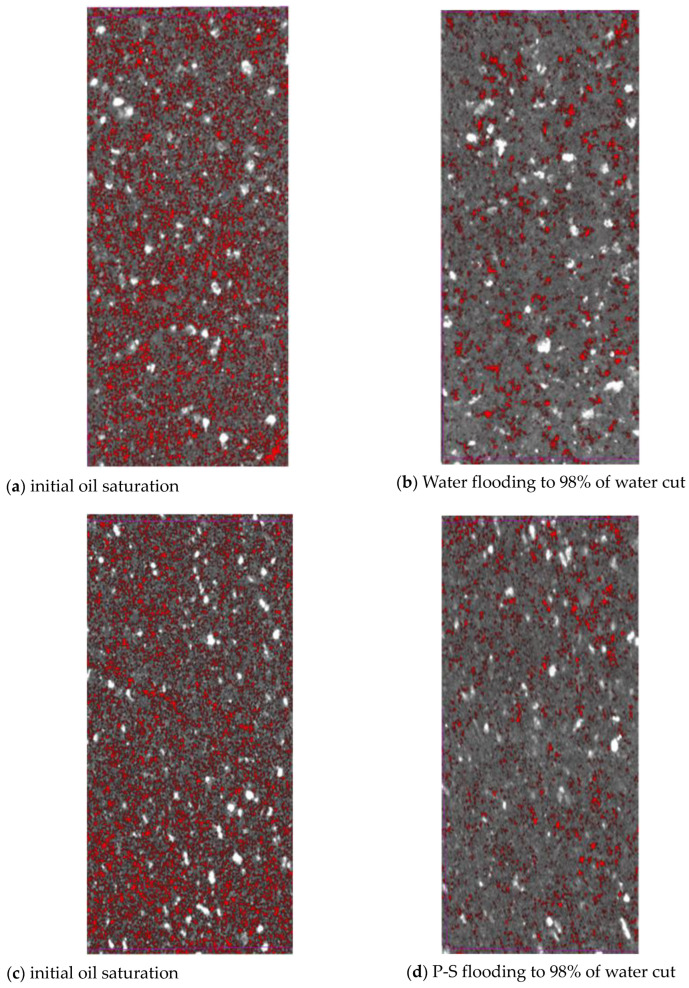
Pore-scale oil distribution after water and P-S flooding of in #2 rock sample.

**Figure 3 molecules-27-08621-f003:**
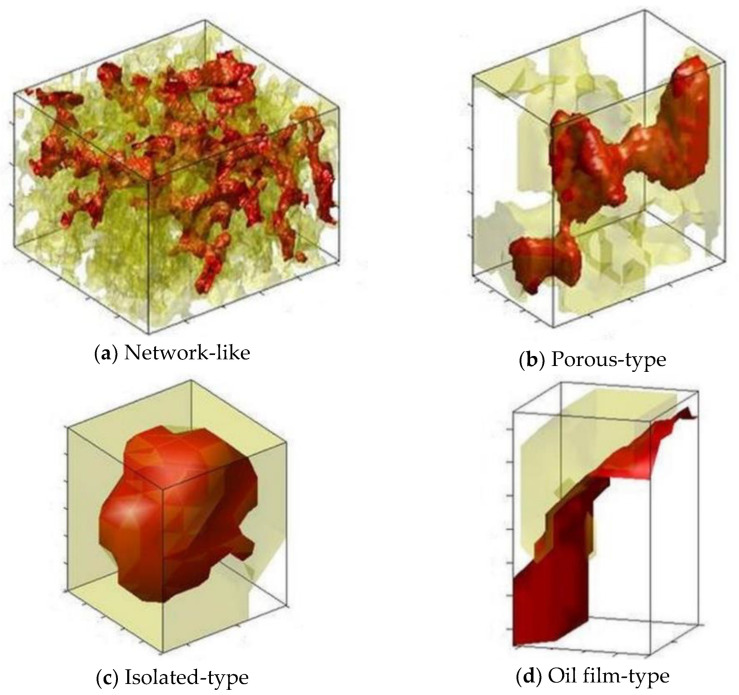
Classification of pore-scale oil morphology in rock samples.

**Figure 4 molecules-27-08621-f004:**
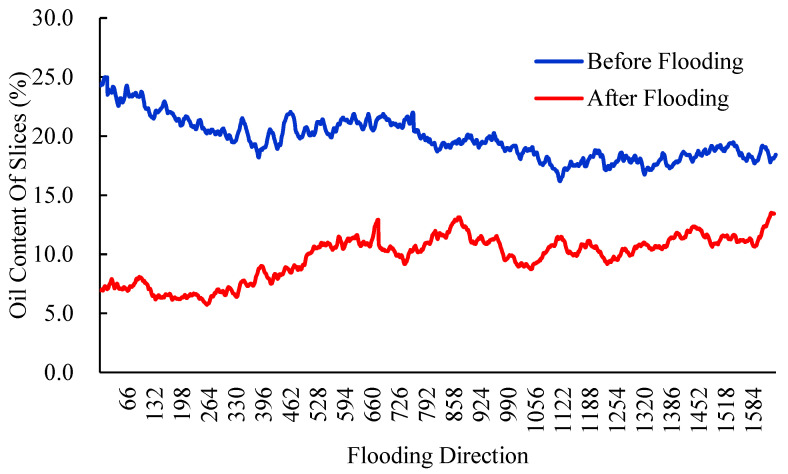
Dynamic change of oil content at different slices along the Z-upward direction.

**Figure 5 molecules-27-08621-f005:**
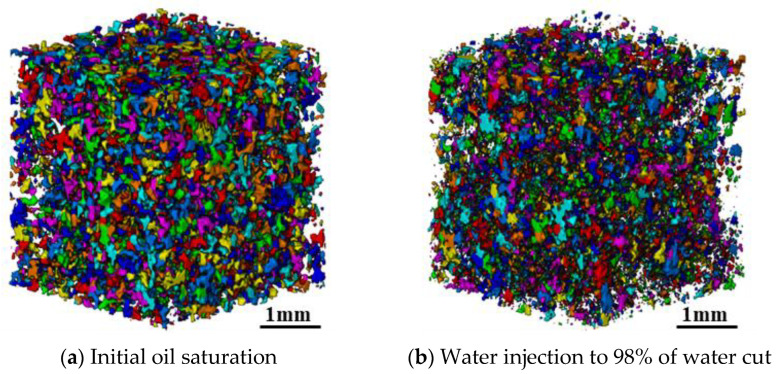
Division of oil droplets during water flooding in the No. 2 rock sample.

**Figure 6 molecules-27-08621-f006:**
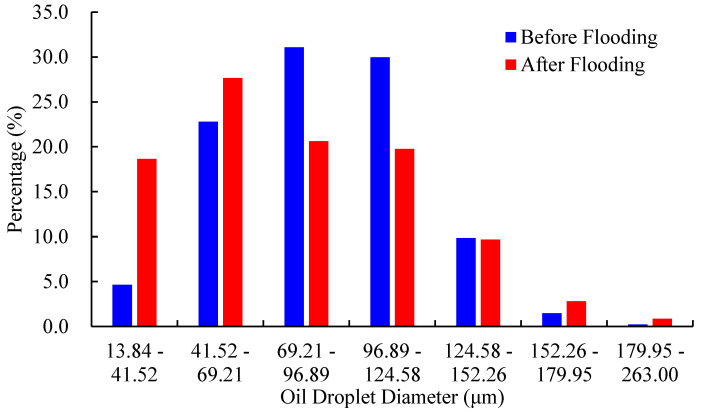
Frequency distribution of oil droplets during water flooding in No. 2 rock sample.

**Figure 7 molecules-27-08621-f007:**
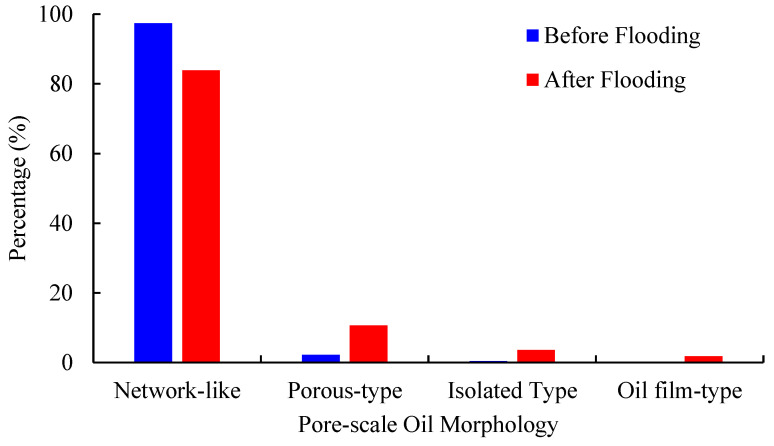
Pore-scale morphology of oil droplets during water flooding in No. 2 rock sample.

**Figure 8 molecules-27-08621-f008:**
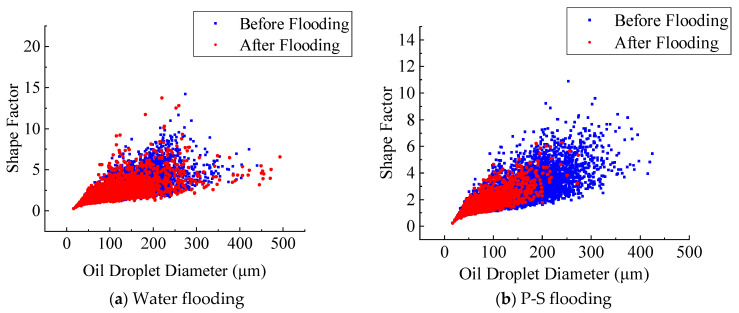
Change in shape factor of residual oil droplets in the No. 1 rock sample.

**Figure 9 molecules-27-08621-f009:**
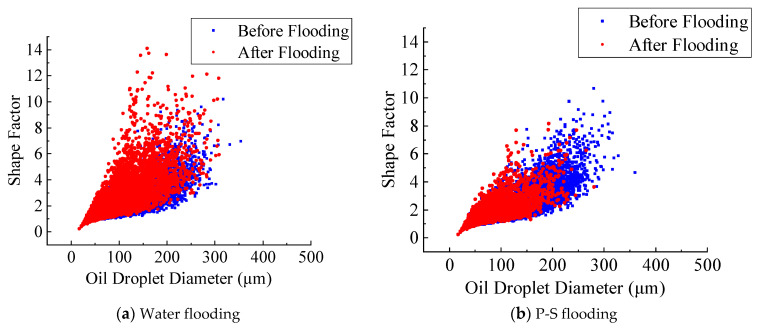
Change in shape factor of residual oil droplets in the No. 2 rock sample.

**Figure 10 molecules-27-08621-f010:**
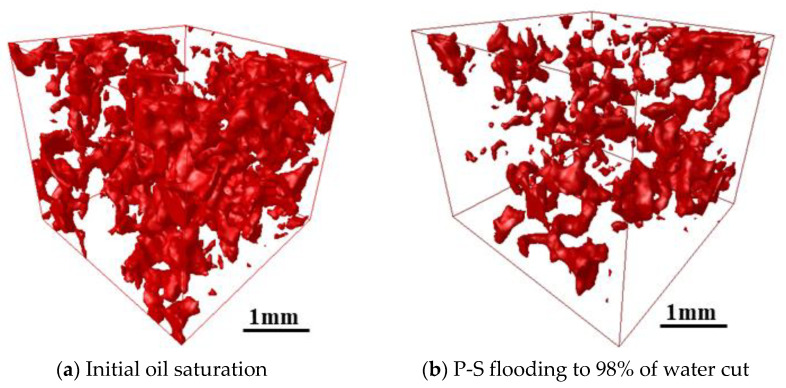
3-D visualization of oil droplet after P-S flooding in the No. 1 rock sample.

**Figure 11 molecules-27-08621-f011:**
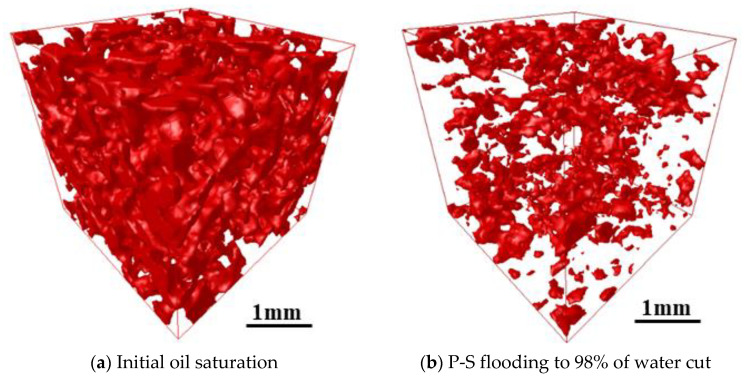
Three-dimensional visualization of oil droplet after P-S flooding in the No. 2 rock sample.

**Figure 12 molecules-27-08621-f012:**
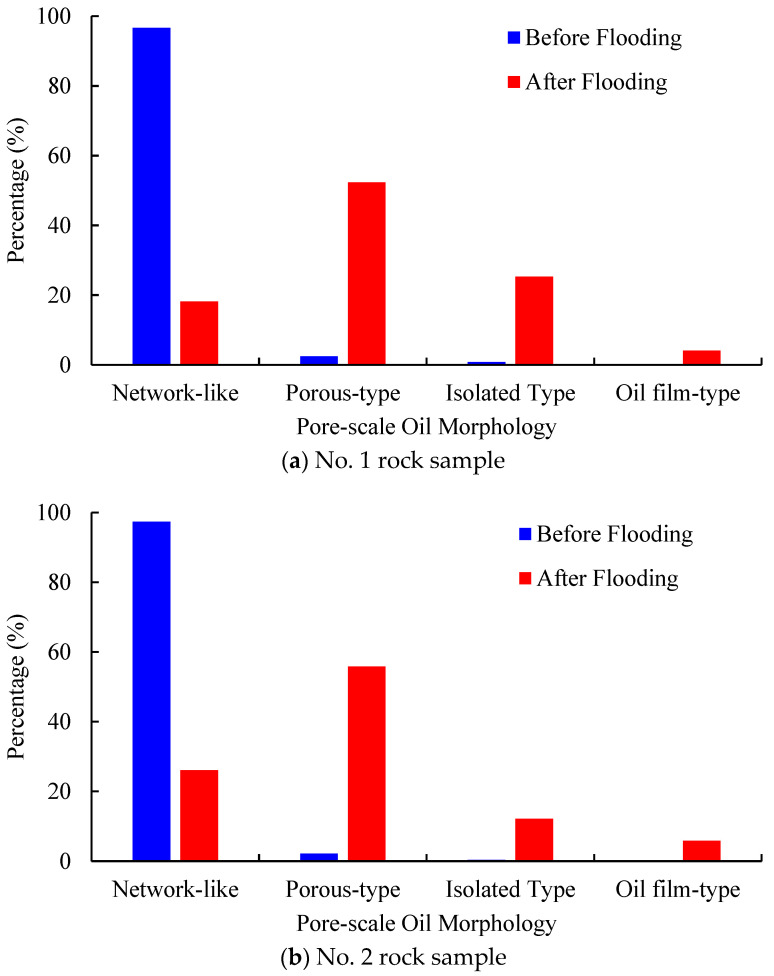
Pore-scale morphology of oil droplets during P-S flooding in both rock samples.

**Figure 13 molecules-27-08621-f013:**
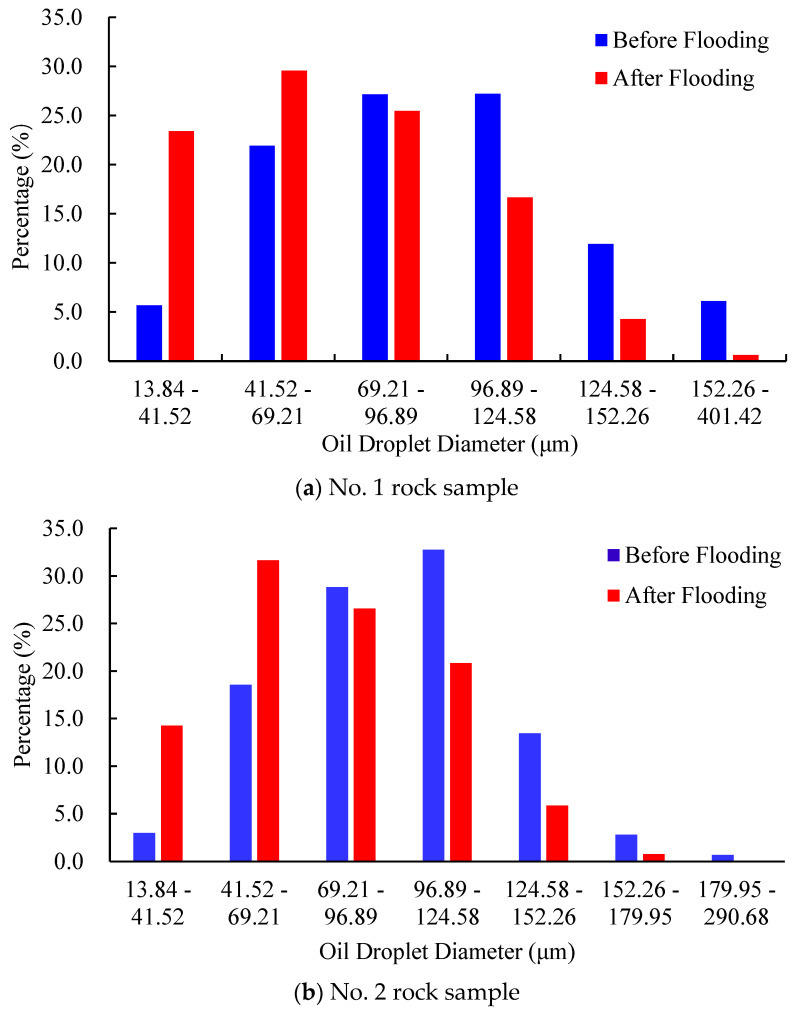
Frequency distribution of oil droplets’ diameter during P-S flooding.

**Table 1 molecules-27-08621-t001:** Core flooding results for the two water-wet rock samples.

No.	Depth (m)	Porosity (%)	Permeability (10^−3^ μm^2^)	Flooding Pattern	Initial Oil Saturation (%)	Residual Oil Saturation (%)	Displacement Efficiency (%)
#1	1067.6	37.1	821.7	Water injection to 98% of water cut	60	26.4	56.0
				Water injection to 90% of water cut, and then P-S flooding to 98% water cut	60	17.1	71.4
#2	1097.3	24.7	646.25	Water injection to 98% of water cut	60	29.5	50.8
				Water injection to 90% of water cut, and then PS flooding to 98% of water cut	60	20.4	65.7

**Table 2 molecules-27-08621-t002:** Geometrical analysis of residual oil droplets in No. 2 rock sample.

CT Imaging Phase	Diameter (μm)	Volume (μm)	Shape Factor	Euler Coefficient	Perimeter (μm)
Initial oil saturation	89.4	1.4 × 10^6^	1.965	−1974	310.7
Water injection to 98% of water cut	79.2	312,702	1.260	−535	80.96

**Table 3 molecules-27-08621-t003:** Average geometric properties of oil droplets during P-S flooding.

No.	CT Imaging Phase	Diameter (μm)	Volume (μm^3^)	Shape Factor	Euler Coefficient	Perimeter (μm)
No. 1	Initial oil saturation	94.4	2.05 × 10^6^	2.08	−1931	406.18
	Displaced to 98% of water cut	69.3	403,262	1.36	697	107.05
No. 2	Initial oil saturation	95.8	1.54 × 10^6^	2.103	−1963	343.4
	Displaced to 98% of water cut	76.1	286,951	1.207	784	76.27

## Data Availability

Detailed data will be provided if requested.

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
