# Peer review of "Study on Oil Recovery Mechanism of Polymer-Surfactant Flooding Using X-ray Microtomography and Integral Geometry"

_molecules, 2022, doi:10.3390/molecules27238621_

Round 1
Reviewer 1 Report
The manuscript is well written with good quality of figures.
-The authors should describe in more detail what is the novelty of their work at the end of the last paragraph in the Introduction section.
-The authors should provide a reference for equation 2.
-Correct '2.1. X-ray micro-CT image processing' to become '3.1 X-ray micro-CT image processing'
-and '2.2. Geometric analysis of oil droplets' to become '3.2. Geometric analysis of oil droplets'
-In line 339 correct summarize to become summarizes
Author Response
[Comment 1]: The authors should describe in more detail what is the novelty of their work at the end of the last paragraph in the Introduction section.
[Reply]: Thank you for your valuable suggestion. We have updated the description to clarify the novelty of our work much clearer at the end of the Introduction section, which is summarized as follows: It is noted that, numerous studies have focused on characterizing the pore-scale morphology and distribution of residual oil in rock samples during waterflooding and their influences on fluid flow, yet few comparative studies on pore-scale oil recovery mechanism of chemical flooding, especially the underlying difference of oil morphology and microstructure between waterflooding and chemical flooding, are performed.
[Comment 2]: The authors should provide a reference for equation 2.
[Reply]: Thank you for your carefully observation. We have added the reference for equation 2 according to the reviewer comment. The modification was colored in Line 298, page 9.
[Comment 3]: Correct '2.1. X-ray micro-CT image processing' to become '3.1 X-ray micro-CT image processing' and '2.2. Geometric analysis of oil droplets' to become '3.2. Geometric analysis of oil droplets'
[Reply]: I feel so sorry for making the small mistake, and we have modified the description carefully according to the reviewer’s suggestion.
[Comment 4]: In line 339 correct summarize to become summarizes
[Reply]: We have corrected the mistakes as the Reviewer suggested, and thank you very much for your careful observation. We also check the whole manuscript in order to diminish the similar errors as much as possible.
Reviewer 2 Report
Dear authors, I received your manuscript for a review.
It seems to me that the article is manuscript interesting and the application of the X-ray CT method for the study of cores of oil-bearing rocks and methods of oil extraction. It is impressive that the authors used the CT method to study essentially large core samples. I think that the article can be published in the Special issue of the Molecules journal. However, a number of changes to the publication are required.
1. The term "integral geometry" is unclear. Did the authors mean joint X-ray tomography experiments and system of pumping of cores with oil-water-surfactants?
2. There are quite a lot of typos and errors in the text. For example "X-CT", "...paris; The ...", "...bijeljic..." and others. Please, check the text of the article carefully.
3. I really liked the authors' Introduction. It is very informative. But it seems redundant to me. There is a lot of information about oil extraction and etc. It seems to me that we should focus more on CT research, microstructure, etc. In any case, the authors have done a lot of work, but this Introduction is good for a large review, in the context of the presented work it is very large and redundant.
4. And again, the term «X-CТ» I meet for the first time. I know CT, X-ray CT, microСТ, muCT. But X-CT - it seems a bit debatable to me...
5. Where are notes on Figure 2. Why are different tomographic slices of sample # 2 presented at the initial state and after flooding? This is indicated by the different location of minerals (light spots) on a slice. Or does the core structure change?
6. P.8, line 267 : “The spatial resolution of a single voxel is 13.84μm.” This is already a repetition. This is stated in the description of the tool.
7. And that the distribution of oil along the core sample is uneven? I calculated 5-7% difference for the original sample for slices 66 and 1584. How can this be explained? That there is more oil left at one end of the sample than at the other? Then, how to scale the experiment to larger or extended core samples?
8. Fig. 5. Please, add a scale bar. For a clearer understanding of the dimensions of the cube. And add a describing of the color scheme. I know for AVIZO it is related to the size of pores and oil droplets. But it is not clear to others, you can add a size scale.
9. Figure 7. “Frequency distribution”. What is measured in? In percentage or number of segmented pores?
10. Why are the «surface area» parameters or «Euler and heterogeneity coefficients» given in Table 2? Are these important parameters? Why then is there no discussion of them in the text of the publication? It seems to me that these parameters are redundant and uninformative...
11. Fig. 10. Please, add a scale bar. For a clearer understanding of the dimensions of the cube.
12. Similarly, in Table 3 in addition to the diameter of the drops and the shape factor, the other «surface area» parameters or «Euler and heterogeneity coefficients» values do not carry a semantic load. Why bring them if there is no discussion of them in the text.
As a result, I want to note that the application of X-ray tomography methods to the study of oil-bearing rocks or cores seems interesting. I am impressed by the creation of an integrated system for the study of flooding processes . But it seems to me that the publication requires changes to improve perception, correct analysis of three-dimensional data and their interpretation.
Author Response
[Comment 1]: The term "integral geometry" is unclear. Did the authors mean joint X-ray tomography experiments and system of pumping of cores with oil-water-surfactants?
[Reply]: I feel so sorry to make the reviewer misunderstand the physical meaning of the term “integral geometry”, a newly developed technique which can be used to quantitatively characterize the pore scale oil morphology and microstructure occupied in the pore space of the rock sample. Using this technique, the topological parameters including Euler coefficient, shape factor, contact angle can be determined, yet as the reviewer described, the geometric analysis of pore-scale oil droplets must originate from high-resolution micro-CT imaging of cores with oil-water-surfactants.
[Comment 2]: There are quite a lot of typos and errors in the text. For example "X-CT", "...paris; The ...", "...bijeljic..." and others. Please, check the text of the article carefully.
[Reply]: Thanks for your valuable suggestion. We have checked and corrected the whole manuscript carefully in order to diminish the typos and errors as the Reviewer described.
[Comment 3]: I really liked the authors' Introduction. It is very informative. But it seems redundant to me. There is a lot of information about oil extraction and etc. It seems to me that we should focus more on CT research, microstructure, etc. In any case, the authors have done a lot of work, but this Introduction is good for a large review, in the context of the presented work it is very large and redundant.
[Reply]: Thank you for your valuable comments. CT research is our topic as the reviewer described, yet we believe that the relevant description about oil extraction and etc is very important to characterize the oil pore morphology and microstructure because accurate segmentation for fluid in pore space typically serves as the basis data to investigate the dynamic evolution of oil morphology and microstructure during chemical flooding.
[Comment 4]: And again, the term «X-CТ» I meet for the first time. I know CT, X-ray CT, microСТ, muCT. But X-CT - it seems a bit debatable to me...
[Reply]: Thank you for your careful observation. We have substituted the term “X-CT” as “X-ray micro-CT” in the revised manuscript.
[Comment 5]: Where are notes on Figure 2. Why are different tomographic slices of sample # 2 presented at the initial state and after flooding? This is indicated by the different location of minerals (light spots) on a slice. Or does the core structure change?
[Reply]: The notes on Figure 2 are listed in Lines 252-263. Due to the experimental operation, it is difficult to keep the rock sample placed in the rotating carrier fixed, so we have to select two different tomographic slices of sample # 2 to show the 2D fluid distribution at the initial state and after flooding. The core structure did not change after waterflooding. The reason for Figure 2 is to demonstrate that as displacement proceeds, oil droplets can be recovered yet there still exists large quantity of remaining oil unexploited from pore space of the rock sample. Even the different tomographic slices are used, the understanding is convinced.
[Comment 6]: P.8, line 267: “The spatial resolution of a single voxel is 13.84μm.” This is already a repetition. This is stated in the description of the tool.
[Reply]: Thank you very much for your careful observation. We have deleted the repeated description as the Reviewer suggested.
[Comment 7]: And that the distribution of oil along the core sample is uneven? I calculated 5-7% difference for the original sample for slices 66 and 1584. How can this be explained? That there is more oil left at one end of the sample than at the other? Then, how to scale the experiment to larger or extended core samples?
[Reply]: As can be seen from Figure 4, the oil content of slices 66 and 1584 differs from each other. Except that the water injection is conducted from one end to the other end of the experimental sample, It is mainly induced by the difference in pore structure between two slices.
[Comment 8]: Fig. 5. Please, add a scale bar. For a clearer understanding of the dimensions of the cube. And add a describing of the color scheme. I know for AVIZO it is related to the size of pores and oil droplets. But it is not clear to others, you can add a size scale.
[Reply]: The description of the color scheme is included in the manuscript, listed in the Line 319, page 10, “Each colored block denotes to a single oil droplet”. The REV is 3.543mm×3.543mm× 3.543mm, which is mentioned above. In order to clarify the description, we have added a scale bar in Fig.5 as the Reviewer suggested, and thanks for your valuable suggestion.
[Comment 9]: Figure 7. “Frequency distribution”. What is measured in? In percentage or number of segmented pores?
[Reply]: It denotes to the percentage. I feel so sorry to puzzle you. We have modified all the Figures to clarify the physical meaning by substituting the “Frequency distribution” with “Percentage (%)”
[Comment 10]: Why are the «surface area» parameters or «Euler and heterogeneity coefficients» given in Table 2? Are these important parameters? Why then is there no discussion of them in the text of the publication? It seems to me that these parameters are redundant and uninformative...
[Reply]: For this study, the surface area is used to calculate the value of shape factor in order to distinguish different pore-scale morphologies of oil droplet. Euler coefficient is used to evaluate the topological connectivity of non-wetting displaced fluid in pore space of the rock samples while the heterogeneity coefficient represents the topological connectivity difference between different oil droplets. Based on the Reviewer’s comment, we delete these parameters to reduce the misunderstanding.
[Comment 11]: Fig. 10. Please, add a scale bar. For a clearer understanding of the dimensions of the cube.
[Reply]: In order to clarify the description, we have added a scale bar in Fig.10 and Fig. 11 as the Reviewer suggested.
[Comment 12]: Similarly, in Table 3 in addition to the diameter of the drops and the shape factor, the other «surface area» parameters or «Euler and heterogeneity coefficients» values do not carry a semantic load. Why bring them if there is no discussion of them in the text.
[Reply]: As above-mentioned, the surface area and Euler heterogeneity coefficients are used as the input data for calculation of shape factor or Euler coefficient to investigate the pore morphology and topology connectivity. Considering the reviewer’s valuable suggestion, we delete these parameters in Table 3.
Round 2
Reviewer 2 Report
The authors have made some changes in accordance with my comments.
The manuscript can be published.